# Optimizing F-Measures by Cost-Sensitive Classification

**Shameem A. Puthiya Parambath,  Nicolas Usunier,  Yves Grandvalet**
Université de Technologie de Compiègne – CNRS, Heudiasyc UMR 7253
Compiègne, France
`{sputhiya,nusunier,grandval}@utc.fr`

## Abstract

We present a theoretical analysis of $F$-measures for binary, multiclass and multilabel classification. These performance measures are non-linear, but in many scenarios they are pseudo-linear functions of the per-class false negative/false positive rate. Based on this observation, we present a general reduction of $F$-measure maximization to cost-sensitive classification with unknown costs. We then propose an algorithm with provable guarantees to obtain an approximately optimal classifier for the $F$-measure by solving a series of cost-sensitive classification problems. The strength of our analysis is to be valid on any dataset and any class of classifiers, extending the existing theoretical results on $F$-measures, which are asymptotic in nature. We present numerical experiments to illustrate the relative importance of cost asymmetry and thresholding when learning linear classifiers on various $F$-measure optimization tasks.

## 1  Introduction

The $F_1$-measure, defined as the harmonic mean of the precision and recall of a binary decision rule [20], is a traditional way of assessing the performance of classifiers. As it favors high and balanced values of precision and recall, this performance metric is usually preferred to (label-dependent weighted) classification accuracy when classes are highly imbalanced and when the cost of a false positive relatively to a false negative is not naturally given for the problem at hand. The design of methods to optimize $F_1$-measure and its variants for multilabel classification (the micro-, macro-, per-instance-$F_1$-measures, see [23] and Section 2), and the theoretical analysis of the optimal classifiers for such metrics have received considerable interest in the last 3-4 years [6, 15, 4, 18, 5, 13], especially because rare classes appear naturally on most multilabel datasets with many labels.

The most usual way of optimizing $F_1$-measure is to perform a two-step approach in which first a classifier which output scores (e.g. a margin-based classifier) is learnt, and then the decision threshold is tuned a posteriori. Such an approach is theoretically grounded in binary classification [15] and for micro- or macro-$F_1$-measures of multilabel classification [13] in that a Bayes-optimal classifier for the corresponding $F_1$-measure can be obtained by thresholding posterior probabilities of classes (the threshold, however, depends on properties of the whole distribution and cannot be known in advance). Thus, such arguments are essentially asymptotic since the validity of the procedure is bound to the ability to accurately estimate all the level sets of the posterior probabilities; in particular, the proof does not hold if one wants to find the optimal classifier for the $F_1$-measure over an arbitrary set of classifiers (e.g. thresholded linear functions).

In this paper, we show that optimizing the $F_1$-measure in binary classification over any (possibly restricted) class of functions and over any data distribution (population-level or on a finite sample) can be reduced to solving an (infinite) series of cost-sensitive classification problems, but the cost space can be discretized to obtain approximately optimal solutions. For binary classification, as well as for multilabel classification (micro-$F_1$-measure in general and the macro-$F_1$-measure when training independent classifiers per class), the discretization can be made along a single real-valued

variable in $[0, 1]$ with approximation guarantees. Asymptotically, our result is, in essence, equivalent to prior results since Bayes-optimal classifiers for cost-sensitive classification are precisely given by thresholding the posterior probabilities, and we recover the relationship between the optimal $F_1$-measure and the optimal threshold given by Lipton et al. [13]. Our reduction to cost-sensitive classification, however, is strictly more general. Our analysis is based on the pseudo-linearity of the $F_1$-scores (the level sets, as function of the false negative rate and the false positive rate are linear) and holds in any asymptotic or non-asymptotic regime, with any arbitrary set of classifiers (without the requirement to output scores or accurate posterior probability estimates). Our formal framework and the definition of pseudo-linearity is presented in the next section, and the reduction to cost-sensitive classification is presented in Section 2.

While our main contribution is the theoretical part, we also turn out to the practical suggestions of our results. In particular, they suggest that, for binary classification, learning cost-sensitive classifiers may be more effective than thresholding probabilities. This is in-line with Musicant et al. [14], although their argument only applies to SVM and does not consider the $F_1$-measure itself but a continuous, non-convex approximation of it. Some experimental results are presented in Section 4, before the conclusion of the paper.

## 2    Pseudo-Linearity and $F$-Measures

Our results are mainly motivated by the maximization of $F$-measures for binary and multilabel classification. They are based on a general property of these performance metrics, namely their pseudo-linearity with respect to the false negative/false positive probabilities.

For binary classification, the results we prove in Section 3 are that in order to optimize the $F$-measure, it is sufficient to solve a binary classification problem with different costs allocated to false positive and false negative errors (Proposition 4). However, these costs are not known *a priori*, so in practice we need to learn several classifiers with different costs, and choose the best one (according to the $F$-score) in a second step. Propositions 5 and 6 provide approximation guarantees on the $F$-score we can obtain by following this principle depending on the granularity of the search in the cost space.

Our results are not specific to the $F_1$-measure in binary classification, and they naturally extend to other cases of $F$-measures with similar functional forms. For that reason, we present the results and prove them directly for the general case, following the framework that we describe in this section. We first present the machine learning framework we consider, and then give the general definition of pseudo-convexity. Then, we provide examples of $F$-measures for binary, multilabel and multiclass classification and we show how they fit into this framework.

### 2.1    Notation and Definitions

We are given *(i)* a measurable space $\mathcal{X} \times \mathcal{Y}$, where $\mathcal{X}$ is the input space and $\mathcal{Y}$ is the (finite) prediction set, *(ii)* a probability measure $\mu$ over $\mathcal{X} \times \mathcal{Y}$, and *(iii)* a set of (measurable) classifiers $\mathcal{H}$ from the input space $\mathcal{X}$ to $\mathcal{Y}$. We distinguish here the prediction set $\mathcal{Y}$ from the label space $\mathcal{L} = \{1, ..., L\}$: in binary or single-label multi-class classification, the prediction set $\mathcal{Y}$ is the label set $\mathcal{L}$, but in multi-label classification, $\mathcal{Y} = 2^{\mathcal{L}}$ is the powerset of the set of possible labels. In that framework, we assume that we have an i.i.d. sample drawn from an underlying data distribution $\mathbb{P}$ on $\mathcal{X} \times \mathcal{Y}$. The empirical distribution of this finite training (or test) sample will be denoted $\hat{\mathbb{P}}$. Then, we may take $\mu = \mathbb{P}$ to get results at the population level (concerning expected errors), or we may take $\mu = \hat{\mathbb{P}}$ to get results on a finite sample. Likewise, $\mathcal{H}$ can be a restricted set of functions such as linear classifiers if $\mathcal{X}$ is a finite-dimensional vector space, or may be the set of all measurable classifiers from $\mathcal{X}$ to $\mathcal{Y}$ to get results in terms of Bayes-optimal predictors. Finally, when needed, we will use bold characters for vectors and normal font with subscript for indexing.

Throughout the paper, we need the notion of pseudo-linearity of a function, which itself is defined from the notion of pseudo-convexity (see e.g. [3, Definition 3.2.1]): a differentiable function $F : \mathcal{D} \subset \mathbb{R}^d \to \mathbb{R}$, defined on a convex open subset of $\mathbb{R}^d$, is *pseudo-convex* if

$$\forall \mathbf{e}, \mathbf{e}' \in \mathcal{D}, \ F(\mathbf{e}) > F(\mathbf{e}') \ \Rightarrow \ \langle \nabla F(\mathbf{e}), \mathbf{e}' - \mathbf{e} \rangle < 0 \ ,$$

where $\langle ., . \rangle$ is the canonical dot product on $\mathbb{R}^d$.

Moreover, $F$ is *pseudo-linear* if both $F$ and $-F$ are pseudo-convex. The important property of pseudo-linear functions is that their level sets are hyperplanes (intersected with the domain), and that sublevel and superlevel sets are half-spaces, all of these hyperplanes being defined by the gradient. In practice, working with gradients of non-linear functions may be cumbersome, so we will use the following characterization, which is a rephrasing of [3, Theorem 3.3.9]:

**Theorem 1 ([3])** *A non-constant function $F : \mathcal{D} \to \mathbb{R}$, defined and differentiable on the open convex set $\mathcal{D} \subseteq \mathbb{R}^d$, is* pseudo-linear *on $\mathcal{D}$ if and only if $\forall \mathbf{e} \in \mathcal{D}$, $\nabla F(\mathbf{e}) \neq \mathbf{0}$, and: $\exists \mathbf{a} : \mathbb{R} \to \mathbb{R}^d$ and $\exists b : \mathbb{R} \to \mathbb{R}$ such that, for any $t$ in the image of $F$:*

$$F(\mathbf{e}) \geq t \iff \langle \mathbf{a}(t), \mathbf{e} \rangle + b(t) \leq 0 \quad and \quad F(\mathbf{e}) \leq t \iff \langle \mathbf{a}(t), \mathbf{e} \rangle + b(t) \geq 0 .$$

Pseudo-linearity is the main property of fractional-linear functions (ratios of linear functions). Indeed, let us consider $F : \mathbf{e} \in \mathbb{R}^d \mapsto (\alpha + \langle \boldsymbol{\beta}, \mathbf{e} \rangle)/(\gamma + \langle \boldsymbol{\delta}, \mathbf{e} \rangle)$ with $\alpha, \gamma \in \mathbb{R}$ and $\boldsymbol{\beta}$ and $\boldsymbol{\delta}$ in $\mathbb{R}^d$. If we restrict the domain of $F$ to the set $\{\mathbf{e} \in \mathbb{R}^d | \gamma + \langle \boldsymbol{\delta}, \mathbf{e} \rangle > 0\}$, then, for all $t$ in the image of $F$ and all $\mathbf{e}$ in its domain, we have: $F(\mathbf{e}) \leq t \iff \langle t\boldsymbol{\delta} - \boldsymbol{\beta}, \mathbf{e} \rangle + t\gamma - \alpha \geq 0$, and the analogous equivalence obtained by reversing the inequalities holds as well; the function thus satisfies the conditions of Theorem 1. As we shall see, many $F$-scores can be written as fractional-linear functions.

## 2.2 Error Profiles and $F$-Measures

For all classification tasks (binary, multiclass and multilabel), the $F$-measures we consider are functions of per-class recall and precision, which themselves are defined in terms of the marginal probabilities of classes and the per-class false negative/false positive probabilities. The marginal probabilities of label $k$ will be denoted by $P_k$, and the per-class false negative/false positive probabilities of a classifier $h$ are denoted by $\mathrm{FN}_k(h)$ and $\mathrm{FP}_k(h)$. Their definitions are given below:

$$(binary/multiclass) \quad P_k = \mu(\{(x, y) | y = k\}), \quad \mathrm{FN}_k(h) = \mu(\{(x, y) | y = k \text{ and } h(x) \neq k\}),$$
$$\mathrm{FP}_k(h) = \mu(\{(x, y) | y \neq k \text{ and } h(x) = k\}).$$

$$(multilabel) \quad P_k = \mu(\{(x, y) | y \in k\}), \quad \mathrm{FN}_k(h) = \mu(\{(x, y) | k \in y \text{ and } k \notin h(x)\}),$$
$$\mathrm{FP}_k(h) = \mu(\{(x, y) | y \notin k \text{ and } k \in h(x)\}).$$

These probabilities of a classifier $h$ are then summarized by the *error profile* $\mathbf{E}(h)$:

$$\mathbf{E}(h) = \big(\mathrm{FN}_1(h), \mathrm{FP}_1(h), ..., \mathrm{FN}_L(h), \mathrm{FP}_L(h)\big) \in \mathbb{R}^{2L} ,$$

so that $e_{2k-1}$ is the false negative probability for class $k$ and $e_{2k}$ is the false positive probability.

**Binary Classification** In binary classification, we have $\mathrm{FN}_2 = \mathrm{FP}_1$ and we write $F$-measures only by reference to class 1. Then, for any $\beta > 0$ and any binary classifier $h$, the $F_\beta$-measure is

$$F_\beta(h) = \frac{(1 + \beta^2)(P_1 - \mathrm{FN}_1(h))}{(1 + \beta^2)P_1 - \mathrm{FN}_1(h) + \mathrm{FP}_1(h)} .$$

The $F_1$-measure, which is the most widely used, corresponds to the case $\beta = 1$. We can immediately notice that $F_\beta$ is fractional-linear, hence pseudo-convex, with respect to $\mathrm{FN}_1$ and $\mathrm{FP}_1$. Thus, with a slight (yet convenient) abuse of notation, we write the $F_\beta$-measure for binary classification as a function of vectors in $\mathbb{R}^4 = \mathbb{R}^{2L}$ which represent error profiles of classifiers:

$$(binary) \quad \forall \mathbf{e} \in \mathbb{R}^4, F_\beta(\mathbf{e}) = \frac{(1 + \beta^2)(P_1 - e_1)}{(1 + \beta^2)P_1 - e_1 + e_2} .$$

**Multilabel Classification** In multilabel classification, there are several definitions of $F$-measures. For those based on the error profiles, we first have the macro-$F$-measures (denoted by $MF_\beta$), which is the average over class labels of the $F_\beta$-measures of each binary classification problem associated to the prediction of the presence/absence of a given class:

$$(multilabel\text{--}Macro) \quad MF_\beta(\mathbf{e}) = \frac{1}{L} \sum_{k=1}^{L} \frac{(1 + \beta^2)(P - e_{2k-1})}{(1 + \beta^2)P - e_{2k-1} + e_{2k}} .$$

$MF_\beta$ *is not* a pseudo-linear function of an error profile $\mathbf{e}$. However, if the multi-label classification algorithm learns independent binary classifiers for each class (a method known as one-vs-rest or binary relevance [23]), then each binary problem becomes independent and optimizing the macro-$F$-score boils down to independently maximizing the $F_\beta$-score for $L$ binary classification problems, so that optimizing $MF_\beta$ is similar to optimizing $F_\beta$ in binary classification.

There are also micro-$F$-measures for multilabel classification. They correspond to $F_\beta$-measures for a new binary classification problem over $\mathcal{X} \times \mathcal{L}$, in which one maps a multilabel classifier $h\colon \mathcal{X} \to \mathcal{Y}$ ($\mathcal{Y}$ is here the power set of $\mathcal{L}$) to the following binary classifier $\tilde{h}\colon \mathcal{X} \times \mathcal{L} \to \{0, 1\}$: we have $\tilde{h}(x, k) = 1$ if $k \in h(x)$, and 0 otherwise. The micro-$F_\beta$-measure, written as a function of an error profile $\mathbf{e}$ and denoted by $mF_\beta(\mathbf{e})$, is the $F_\beta$-score of $\tilde{h}$ and can be written as:

$$(multilabel-micro) \qquad mF_\beta(\mathbf{e}) = \frac{(1 + \beta^2) \sum_{k=1}^{L} (P_k - e_{2k-1})}{(1 + \beta^2) \sum_{k=1}^{L} P_k + \sum_{k=1}^{L} (e_{2k} - e_{2k-1})} .$$

This function is also fractional-linear, and thus pseudo-linear as a function of $\mathbf{e}$.

A third notion of $F_\beta$-measure can be used in multilabel classification, namely the per-instance $F_\beta$ studied e.g. by [16, 17, 6, 4, 5]. The per-instance $F_\beta$ is defined as the average, over instances $x$, of the binary $F_\beta$-measure for the problem of classifying labels given $x$. This corresponds to a specific $F_\beta$-maximization problem for each $x$ and is not directly captured by our framework, because we would need to solve different cost-sensitive classification problems for each instance.

**Multiclass Classification**   The last example we take is from multiclass classification. It differs from multilabel classification in that a single class must be predicted for each example. This restriction imposes strong global constraints that make the task significantly harder. As for the multillabel case, there are many definitions of $F$-measures for multiclass classification, and in fact several definitions for the micro-$F$-measure itself. We will focus on the following one, which is used in information extraction (e.g. in the BioNLP challenge [12]). Given $L$ class labels, we will assume that label 1 corresponds to a "default" class, the prediction of which is considered as not important. In information extraction, the "default" class corresponds to the (majority) case where no information should be extracted. Then, a false negative is an example $(x, y)$ such that $y \neq 1$ and $h(x) \neq y$, while a false positive is an example $(x, y)$ such that $y = 1$ and $h(x) \neq y$. This micro-$F$-measure, denoted $mcF_\beta$ can be written as:

$$(multiclass-micro) \qquad mcF_\beta(\mathbf{e}) = \frac{(1 + \beta^2)(1 - P_1 - \sum_{k=2}^{L} e_{2k-1})}{(1 + \beta^2)(1 - P_1) - \sum_{k=2}^{L} e_{2k-1} + e_1} .$$

Once again, this kind of micro-$F_\beta$-measure is pseudo-linear with respect to $\mathbf{e}$.

**Remark 2 (Training and generalization performance)** *Our results concern a fixed distribution $\mu$, while the goal is to find a classifier with high generalization performance. With our notation, our results apply to $\mu = \mathbb{P}$ or $\mu = \hat{\mathbb{P}}$, and our implicit goal is to perform empirical risk minimization-type learning, that is, to find a classifier with high value of $F_\beta^{\mathbb{P}}\left(\mathbf{E}^{\mathbb{P}}(h)\right)$ by maximizing its empirical counterpart $F_\beta^{\hat{\mathbb{P}}}\left(\mathbf{E}^{\hat{\mathbb{P}}}(h)\right)$ (the superscripts here make the underlying distribution explicit).*

**Remark 3 (Expected Utility Maximization (EUM) vs Decision-Theoretic Approach (DTA))**
*Nan et al. [15] propose two possible definitions of the generalization performance in terms of $F_\beta$-scores. In the first framework, called EUM, the population-level $F_\beta$-score is defined as the $F_\beta$-score of the population-level error profiles. In contrast, the Decision-Theoretic approach defines the population-level $F_\beta$-score as the expected value of the $F_\beta$-score over the distribution of test sets. The EUM definition of generalization performance matches our framework using $\mu = \mathbb{P}$: in that sense, we follow the EUM framework. Nonetheless, regardless of how we define the generalization performance, our results can be used to maximize the empirical value of the $F_\beta$-score.*

## 3   Optimizing $F$-Measures by Reduction to Cost-Sensitive Classification

The $F$-measures presented above are non-linear aggregations of false negative/positive probabilities that cannot be written in the usual expected loss minimization framework; usual learning algorithms are thus, intrinsically, not designed to optimize this kind of performance metrics.

In this section, we show in Proposition 4 that the optimal classifier for a cost-sensitive classification problem with label dependent costs [7, 24] is also an optimal classifier for the pseudo-linear $F$-measures (within a specific, yet arbitrary classifier set $\mathcal{H}$). In cost-sensitive classification, each entry of the error profile is weighted by a non-negative cost, and the goal is to minimize the weighted average error. Efficient, consistent algorithms exist for such cost-sensitive problems [1, 22, 21]. Even though the costs corresponding to the optimal $F$-score are not known *a priori*, we show in Proposition 5 that we can approximate the optimal classifier with approximate costs. These costs, explicitly expressed in terms of the optimal $F$-score, motivate a practical algorithm.

### 3.1   Reduction to Cost-Sensitive Classification

In this section, $F : \mathcal{D} \subset \mathbb{R}^d \to \mathbb{R}$ is a fixed pseudo-linear function. We denote by $\mathbf{a} : \mathbb{R} \to \mathbb{R}^d$ the function mapping values of $F$ to the corresponding hyperplane of Theorem 1. We assume that the distribution $\mu$ is fixed, as well as the (arbitrary) set of classifier $\mathcal{H}$. We denote by $\mathcal{E}(\mathcal{H})$ the closure of the image of $\mathcal{H}$ under $\mathbf{E}$, i.e. $\mathcal{E}(\mathcal{H}) = cl(\{\mathbf{E}(h), h \in \mathcal{H}\})$ (the closure ensures that $\mathcal{E}(\mathcal{H})$ is compact and that minima/maxima are well-defined), and we assume $\mathcal{E}(\mathcal{H}) \subseteq \mathcal{D}$. Finally, for the sake of discussion with cost-sensitive classification, we assume that $\mathbf{a}(t) \in \mathbb{R}_+^d$ for any $\mathbf{e} \in \mathcal{E}(\mathcal{H})$, that is, lower values of errors entail higher values of $F$.

**Proposition 4** *Let $F^\star = \max\limits_{\mathbf{e}' \in \mathcal{E}(\mathcal{H})} F(\mathbf{e}')$. We have:* $\mathbf{e} \in \operatorname*{argmin}\limits_{\mathbf{e}' \in \mathcal{E}(\mathcal{H})} \langle \mathbf{a}(F^\star), \mathbf{e}' \rangle \;\Leftrightarrow\; F(\mathbf{e}) = F^\star$

*Proof* Let $\mathbf{e}^\star \in \operatorname{argmax}_{\mathbf{e}' \in \mathcal{E}(\mathcal{H})} F(\mathbf{e}')$, and let $\mathbf{a}^\star = \mathbf{a}(F(\mathbf{e}^\star)) = \mathbf{a}(F^\star)$. We first notice that pseudo-linearity implies that the set of $\mathbf{e} \in \mathcal{D}$ such that $\langle \mathbf{a}^\star, \mathbf{e} \rangle = \langle \mathbf{a}^\star, \mathbf{e}^\star \rangle$ corresponds to the level set $\{\mathbf{e} \in \mathcal{D} | F(\mathbf{e}) = F(\mathbf{e}^\star) = F^\star\}$. Thus, we only need to show that $\mathbf{e}^\star$ is a minimizer of $\mathbf{e}' \mapsto \langle \mathbf{a}^\star, \mathbf{e}' \rangle$ in $\mathcal{E}(\mathcal{H})$. To see this, we notice that pseudo-linearity implies

$$\forall \mathbf{e}' \in \mathcal{D}, \; F(\mathbf{e}^\star) \geq F(\mathbf{e}') \Rightarrow \langle \mathbf{a}^\star, \mathbf{e}^\star \rangle \leq \langle \mathbf{a}^\star, \mathbf{e}' \rangle$$

from which we immediately get $\mathbf{e}^\star \in \operatorname{argmin}_{\mathbf{e}' \in \mathcal{E}(\mathcal{H})} \langle \mathbf{a}^\star, \mathbf{e}' \rangle$ since $\mathbf{e}^\star$ maximizes $F$ in $\mathcal{E}(\mathcal{H})$.   $\square$

The proposition shows that $\mathbf{a}(F^\star)$ are the costs that should be assigned to the error profile in order to find the $F$-optimal classifier in $\mathcal{H}$. Hence maximizing $F$ amounts to minimizing $\langle \mathbf{a}(F^\star), \mathbf{E}(h) \rangle$ with respect to $h$, that is, amounts to solving a cost-sensitive classification problem. The costs $\mathbf{a}(F^\star)$ are, however, not known *a priori* (because $F^\star$ is not known in general). The following result shows that having only approximate costs is sufficient to have an approximately $F$-optimal solution, which gives us the main step towards a practical solution:

**Proposition 5** *Let $\varepsilon_0 \geq 0$ and $\varepsilon_1 \geq 0$, and assume that there exists $\Phi > 0$ such that for all $\mathbf{e}, \mathbf{e}' \in \mathcal{E}(\mathcal{H})$ satisfying $F(\mathbf{e}') > F(\mathbf{e})$, we have:*

$$F(\mathbf{e}') - F(\mathbf{e}) \leq \Phi \langle \mathbf{a}(F(\mathbf{e}')), \mathbf{e} - \mathbf{e}' \rangle . \tag{1}$$

*Then, let us take $\mathbf{e}^\star \in \operatorname{argmax}_{\mathbf{e}' \in \mathcal{E}(\mathcal{H})} F(\mathbf{e}')$, and denote $\mathbf{a}^\star = \mathbf{a}(F(\mathbf{e}^\star))$. Let furthermore $\mathbf{g} \in \mathbb{R}_+^d$ and $h \in \mathcal{H}$ satisfying the two following conditions:*

*(i)* $\| \mathbf{g} - \mathbf{a}^\star \|_2 \leq \varepsilon_0$          *(ii)* $\langle \mathbf{g}, \mathbf{E}(h) \rangle \leq \min\limits_{\mathbf{e}' \in \mathcal{E}(\mathcal{H})} \langle \mathbf{g}, \mathbf{e}' \rangle + \varepsilon_1 .$

*We have:*      $F(\mathbf{E}(h)) \geq F(\mathbf{e}^\star) - \Phi \cdot (2\varepsilon_0 M + \varepsilon_1)$, *where* $M = \max\limits_{\mathbf{e}' \in \mathcal{E}(\mathcal{H})} \| \mathbf{e}' \|_2.$

*Proof* Let $\mathbf{e}' \in \mathcal{E}(\mathcal{H})$. By writing $\langle \mathbf{g}, \mathbf{e}' \rangle = \langle \mathbf{g} - \mathbf{a}^\star, \mathbf{e}' \rangle + \langle \mathbf{a}^\star, \mathbf{e}' \rangle$ and applying Cauchy-Schwarz inequality to $\langle \mathbf{g} - \mathbf{a}^\star, \mathbf{e}' \rangle$ we get $\langle \mathbf{g}, \mathbf{e}' \rangle \leq \langle \mathbf{a}^\star, \mathbf{e}' \rangle + \varepsilon_0 M$ using condition *(i)*. Consequently

$$\min\limits_{\mathbf{e}' \in \mathcal{E}(\mathcal{H})} \langle \mathbf{g}, \mathbf{e}' \rangle \leq \min\limits_{\mathbf{e}' \in \mathcal{E}(\mathcal{H})} \langle \mathbf{a}^\star, \mathbf{e}' \rangle + \varepsilon_0 M = \langle \mathbf{a}^\star, \mathbf{e}^\star \rangle + \varepsilon_0 M \tag{2}$$

Where the equality is given by Proposition 4. Now, let $\mathbf{e} = \mathbf{E}(h)$, assuming that classifier $h$ satisfies condition *(ii)*. Using $\langle \mathbf{a}^\star, \mathbf{e} \rangle = \langle \mathbf{a}^\star - \mathbf{g}, \mathbf{e} \rangle + \langle \mathbf{g}, \mathbf{e} \rangle$ and Cauchy-Shwarz, we obtain:

$$\langle \mathbf{a}^\star, \mathbf{e} \rangle \leq \langle \mathbf{g}, \mathbf{e} \rangle + \varepsilon_0 M \leq \min\limits_{\mathbf{e}' \in \mathcal{E}(\mathcal{H})} \langle \mathbf{g}, \mathbf{e}' \rangle + \varepsilon_1 + \varepsilon_0 M \leq \langle \mathbf{a}^\star, \mathbf{e}^\star \rangle + \varepsilon_1 + 2\varepsilon_0 M ,$$

where the first inequality comes from condition *(ii)* and the second inequality comes from (2). The final result is obtained by plugging this inequality into (1).   $\square$

Before discussing this result, we first give explicit values of **a** and $\Phi$ for pseudo-linear $F$-measures:

**Proposition 6** $F_\beta$, $mF_\beta$ and $mcF_\beta$ defined in Section 2 satisfy the conditions of Proposition 5 with:

$(binary)$ $F_\beta$: $\qquad\qquad \Phi = \dfrac{1}{\beta^2 P_1} \qquad\qquad$ and $\mathbf{a} : t \in [0,1] \mapsto (1+\beta^2 - t, t, 0, 0)$.

$(multilabel-micro)$ $mF_\beta$: $\quad \Phi = \dfrac{1}{\beta^2 \sum_{k=1}^{L} P_k} \quad$ and $a_i(t) = \begin{cases} 1 + \beta^2 - t & \text{if } i \text{ is odd} \\ t & \text{if } i \text{ is even} \end{cases}$.

$(multiclass-micro)$ $mcF_\beta$: $\quad \Phi = \dfrac{1}{\beta^2(1-P_1)} \quad$ and $a_i(t) = \begin{cases} 1 + \beta^2 - t & \text{if } i \text{ is odd and } i \neq 1 \\ t & \text{if } i = 1 \\ 0 & \text{otherwise} \end{cases}$.

The proof is given in the longer version of the paper, and the values of $\Phi$ and **a** are valid for any set of classifiers $\mathcal{H}$. Note that the result on $F_\beta$ for binary classification can be used for the macro-$F_\beta$-measure in multilabel classification when training one binary classifier per label. Also, the relative costs $(1+\beta^2-t)$ for false negative and $t$ for false positive imply that for the $F_1$-measure, the optimal classifier is the solution of the cost-sensitive binary problem with costs $(1 - F^\star/2)$, $F^\star/2$. If we take $\mathcal{H}$ as the set of all measurable functions, the Bayes-optimal classifier for this cost is to predict class 1 when $\mu(y = 1|x) \geq F^\star/2$ (see e.g. [22]). Our propositions thus extends this known result [13] to the non-asymptotic regime and to an arbitrary set of classifiers.

## 3.2 Practical Algorithm

Our results suggests that the optimization of pseudo-linear $F$-measures should wrap cost-sensitive classification algorithms, used in an inner loop, by an outer loop setting the appropriate costs. In practice, since the function $\mathbf{a} : [0,1] \to \mathbb{R}^d$, which assigns costs to probabilities of error, is Lipschitz-continuous (with constant 2 on our examples), it is sufficient to discretize the interval $[0,1]$ to have a set of evenly spaced values $\{t_1, ..., t_C\}$ (say, $t_{j+1} - t_j = \varepsilon_0/2$) to obtain an $\varepsilon_0$-cover $\{\mathbf{a}(t_1), ..., \mathbf{a}(t_C)\}$ of the possible costs. Using the approximate guarantee of Proposition 5, learning a cost-sensitive classifier for each $\mathbf{a}(t_i)$ and selecting the one with optimal $F$-measure *a posteriori* is sufficient to obtain a $M\Phi(2\varepsilon_0 + \varepsilon_1)$-optimal solution, where $\varepsilon_1$ is the approximation guarantee of the cost-sensitive classification algorithm.

This meta-algorithm can be instantiated with any learning algorithm and different $F$-measures. In our experiments of Section 4, we first use it with cost-sensitive binary classification algorithms: Support Vector Machines (SVMs) and logistic regression, both with asymmetric costs [2], to optimize the $F_1$-measure in binary classification and the macro-$F_1$-score in multilabel classification (training one-vs-rest classifiers). Musicant et al. [14] also advocated for SVMs with asymmetric costs for $F_1$-measure optimization in binary classification. However, their argument, specific to SVMs, is not methodological but technical (relaxation of the maximization problem).

## 4 Experiments

The goal of this section is to give illustration of the algorithms suggested by the theory. First, our results suggest that cost-sensitive classification algorithms may be preferable to the more usual probability thresholding method. We compare cost-sensitive classification, as implemented by SVMs with asymmetric costs, to thresholded logistic regression, with linear classifiers. Besides, the structured SVM approach to $F_1$-measure maximization SVM$^{\text{perf}}$ [11] provides another baseline. For completeness, we also report results for thresholded SVMs, cost-sensitive logistic regression, and for the thresholded versions of SVM$^{\text{perf}}$ and the cost-sensitive algorithms (a *thresholded* algorithm means that the decision threshold is tuned *a posteriori* by maximizing the $F_1$-score on the validation set).

Cost-sensitive SVMs and logistic regression (LR) differ in the loss they optimize (weighted hinge loss for SVMs, weighted log-loss for LR), and even though both losses are calibrated in the cost-sensitive setting (that is, converging toward a Bayes-optimal classifier as the number of examples and the capacity of the class of function grow to infinity) [22], they behave differently on finite datasets or with restricted classes of functions. We may also note that asymptotically, the Bayes-classifier for

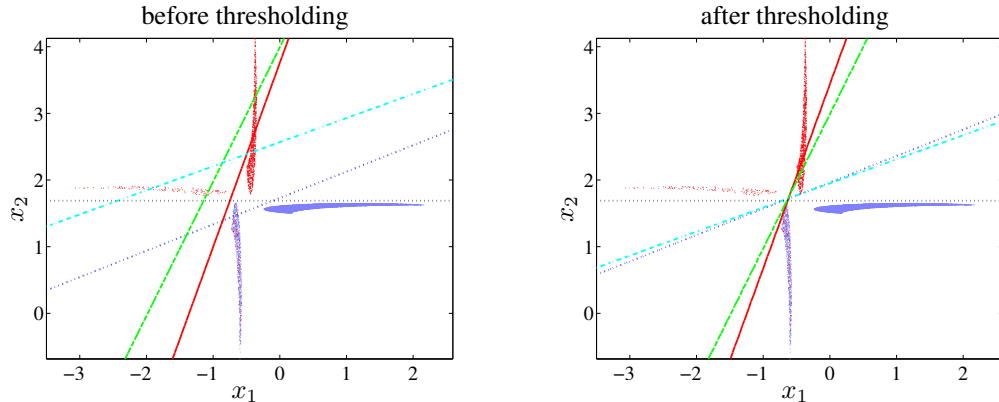

Figure 1: Decision boundaries for the galaxy dataset before and after thresholding the classifier scores of SVM[perf] (dotted, blue), cost-sensitive SVM (dot-dashed, cyan), logistic regression (solid, red), and cost-sensitive logistic regression (dashed, green). The horizontal black dotted line is an optimal decision boundary.

a cost-sensitive binary classification problem is a classifier which thresholds the posterior probability of being class 1. Thus, all methods but SVM[perf] are asymptotically equivalent, and our goal here is to analyze their non-asymptotic behavior on a restricted class of functions.

Although our theoretical developments do not indicate any need to threshold the scores of classifiers, the practical benefits of a post-hoc adjustment of these scores can be important in terms of $F_1$-measure maximization. The reason is that the decision threshold given by cost-sensitive SVMs or logistic regression might not be optimal in terms of the cost-sensitive 0/1-error, as already noted in cost-sensitive learning scenarios [10, 2]. This is illustrated in Figure 1, on the didactic "Galaxy" distribution, consisting in four clusters of 2D-examples, indexed by $z \in \{1, 2, 3, 4\}$, with prior probability $\mathbb{P}(z = 1) = 0.01$, $\mathbb{P}(z = 2) = 0.1$, $\mathbb{P}(z = 3) = 0.001$, and $\mathbb{P}(z = 4) = 0.889$, with respective class conditional probabilities $\mathbb{P}(y = 1|z = 1) = 0.9$, $\mathbb{P}(y = 1|z = 2) = 0.09$, $\mathbb{P}(y = 1|z = 3) = 0.9$, and $\mathbb{P}(y = 1|z = 4) = 0$. We drew a very large sample (100,000 examples) from the distribution, whose optimal $F_1$-measure is 67.5%. Without tuning the decision threshold of the classifiers, the best $F_1$-measure among the classifiers is 55.3%, obtained by SVM[perf], whereas tuning thresholds enables to reach the optimal $F_1$-measure for SVM[perf] and cost-sensitive SVM. On the other hand, LR is severely affected by the non-linearity of the level sets of the posterior probability distribution, and does not reach this limit (best $F_1$-score of 48.9%). Note also that even with this very large sample size, the SVM and LR classifiers are very different.

The datasets we use are `Adult` (binary classification, 32,561/16,281 train/test ex., 123 features), `Letter` (single label multiclass, 26 classes, 20,000 ex., 16 features), and two text datasets: the 20 Newsgroups dataset `News20`[1] (single label multiclass, 20 classes, 15,935/3,993 train/test ex., 62,061 features, scaled version) and `Siam`[2] (multilabel, 22 classes, 21,519/7,077 train/test ex., 30,438 features). All datasets except for `News20` and `Siam` are obtained from the UCI repository[3]. For each experiment, the training set was split at random, keeping 1/3 for the validation set used to select all hyper-parameters, based on the maximization of the $F_1$-measure on this set. For datasets that do not come with a separate test set, the data was first split to keep 1/4 for test. The algorithms have from one to three hyper-parameters: *(i)* all algorithms are run with $L_2$ regularization, with a regularization parameter $C \in \{2^{-6}, 2^{-5}, ..., 2^6\}$; *(ii)* for the cost-sensitive algorithms, the cost for false negatives is chosen in $\{\frac{2-t}{t}, t \in \{0.1, 0.2, ..., 1.9\}\}$ of Proposition 6[4]; *(iii)* for the thresholded algorithms, the threshold is chosen among all the scores of the validation examples.

Table 1: (macro-)$F_1$-measures (in %). Options: T stands for thresholded, CS for cost-sensitive and CS&T for cost-sensitive and thresholded.

| Baseline Options | SVM$^{perf}$ – | SVM$^{perf}$ T | SVM T | SVM CS | SVM CS&T | LR T | LR CS | LR CS&T |
|---|---|---|---|---|---|---|---|---|
| `Adult` | 67.3 | **67.9** | 67.8 | **67.9** | 67.8 | 67.8 | **67.9** | 67.8 |
| `Letter` | 52.5 | 60.8 | 63.1 | 63.2 | **63.8** | 61.2 | 59.9 | 62.1 |
| `News20` | 59.5 | 78.7 | 82.0 | 81.7 | **82.4** | 81.2 | 81.1 | 81.5 |
| `Siam` | 49.4 | 52.8 | 52.6 | 51.9 | **54.9** | 53.9 | 53.8 | 54.4 |

The library *LibLinear* [9] was used to implement SVMs[5] and Logistic Regression (LR). A constant feature with value 100 was added to each dataset to mimic an unregularized offset.

The results, averaged over five random splits, are reported in Table 1. As expected, the difference between methods is less extreme than on the artificial "Galaxy" dataset. The `Adult` dataset is an example where all methods perform nearly identically; the surrogate loss used in practice seems unimportant. On the other datasets, we observe that thresholding has a rather large impact, and especially for SVM$^{perf}$; this is also true for the other classifiers: the unthresholded SVM and LR with symmetric costs (unreported here) were not competitive as well. The cost-sensitive (thresholded) SVM outperforms all other methods, as suggested by the theory. It is probably the method of choice when predictive performance is a must.

On these datasets, thresholded LR behaves reasonably well considering its relatively low computational cost. Indeed, LR is much faster than SVM: in their thresholded cost-sensitive versions, the timings for LR on `News20` and `Siam` datasets are 6,400 and 8,100 seconds, versus 255,000 and 147,000 seconds for SVM respectively. Note that we did not try to optimize the running time in our experiments. In particular, considerable time savings could be achieved by using warm-start.

# 5   Conclusion

We presented an analysis of $F$-measures, leveraging the property of pseudo-linearity of some of them to obtain a strong non-asymptotic reduction to cost-sensitive classification. The results hold for any dataset and for any class of function. Our experiments on linear functions confirm theory, by demonstrating the practical interest of using cost-sensitive classification algorithms rather than using a simple probability thresholding. However, they also reveal that, for $F$-measure maximization, thresholding the solutions provided by cost-sensitive algorithms further improves performances.

Algorithmically and empirically, we only explored the simplest case of our result ($F_\beta$-measure in binary classification and macro-$F_\beta$-measure in multilabel classification), but much more remains to be done. First, the strategy we use for searching the optimal costs is a simple uniform discretization procedure, and more efficient exploration techniques could probably be developed. Second, algorithms for the optimization of the micro-$F_\beta$-measure in multilabel classification received interest recently as well [8, 19], but are for now limited to the selection of threshold after any kind of training. New methods for that measure may be designed from our reduction; we also believe that our result can lead to progresses towards optimizing the micro-$F_\beta$ measure in multiclass classification.

**Acknowledgments**

This work was carried out and funded in the framework of the Labex MS2T. It was supported by the Picardy Region and the French Government, through the program "Investments for the future" managed by the National Agency for Research (Reference ANR-11-IDEX-0004-02).

## Footnotes

[1] http://www.csie.ntu.edu.tw/~cjlin/libsvmtools/datasets/multiclass.html#news20

[2] http://www.csie.ntu.edu.tw/~cjlin/libsvmtools/datasets/multilabel.html#siam-competition2007

[3] https://archive.ics.uci.edu/ml/datasets.html

[4] We take $t$ greater than 1 in case the training asymmetry would be different from the true asymmetry [2].

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
