[Supplementary Material]

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

 $\left\langle \mathbf{a}^\star, \mathbf{e}\right\rangle = \left\langle \mathbf{a}^\star, \mathbf{e}^\star\right\rangle$ corresponds to the level set $\{\mathbf{e} \in \mathcal{D} | F(\mathbf{e}) = F(\mathbf{e}^\star) = F^\star\}$. Thus, we only need to show that $\mathbf{e}^\star$ is a minimizer of $\mathbf{e}' \mapsto \left\langle \mathbf{a}^\star, \mathbf{e}'\right\rangle$ in $\mathcal{E}\left(\mathcal{H}\right)$. To see this, we notice that pseudo-linearity implies

$$\forall \mathbf{e}' \in \mathcal{D}, \; F(\mathbf{e}^\star) \geq F(\mathbf{e}') \Rightarrow \left\langle \mathbf{a}^\star, \mathbf{e}^\star\right\rangle \leq \left\langle \mathbf{a}^\star, \mathbf{e}'\right\rangle$$

from which we immediately get $\mathbf{e}^\star \in \operatorname{argmin}_{\mathbf{e}' \in \mathcal{E}(\mathcal{H})} \left\langle \mathbf{a}^\star, \mathbf{e}'\right\rangle$ since $\mathbf{e}^\star$ maximizes $F$ in $\mathcal{E}\left(\mathcal{H}\right)$. $\quad\square$

The proposition shows that $\mathbf{a}\left(F^\star\right)$ are the costs that should be assigned to the error profile in order to find the $F$-optimal classifier in $\mathcal{H}$. Hence maximizing $F$ amounts to minimizing $\left\langle \mathbf{a}\left(F^\star\right), \mathbf{E}(h)\right\rangle$ with respect to $h$, that is, amounts to solving a cost-sensitive classification problem. The costs $\mathbf{a}\left(F^\star\right)$ are, however, not known *a priori* (because $F^\star$ is not known in general). The following result shows that having only approximate costs is sufficient to have an approximately $F$-optimal solution, which gives us the main step towards a practical solution:

**Proposition 5** *Let $\varepsilon_0 \geq 0$ and $\varepsilon_1 \geq 0$, and assume that there exists $\Phi > 0$ such that for all $\mathbf{e}, \mathbf{e}' \in \mathcal{E}\left(\mathcal{H}\right)$ satisfying $F(\mathbf{e}') > F(\mathbf{e})$, we have:*

$$F(\mathbf{e}') - F(\mathbf{e}) \leq \Phi \left\langle \mathbf{a}(F(\mathbf{e}')), \mathbf{e} - \mathbf{e}'\right\rangle . \tag{1}$$

*Then, let us take $\mathbf{e}^\star \in \operatorname{argmax}_{\mathbf{e}' \in \mathcal{E}(\mathcal{H})} F(\mathbf{e}')$, and denote $\mathbf{a}^\star = \mathbf{a}(F(\mathbf{e}^\star))$. Let furthermore $\mathbf{g} \in \mathbb{R}^d_+$ and $h \in \mathcal{H}$ satisfying the two following conditions:*

*(i)* $\| \mathbf{g} - \mathbf{a}^\star \|_2 \leq \varepsilon_0$   *(ii)* $\left\langle \mathbf{g}, \mathbf{E}(h)\right\rangle \leq \min\limits_{\mathbf{e}' \in \mathcal{E}(\mathcal{H})} \left\langle \mathbf{g}, \mathbf{e}'\right\rangle + \varepsilon_1$ .

*We have:* $F(\mathbf{E}(h)) \geq F(\mathbf{e}^\star) - \Phi \cdot (2\varepsilon_0 M + \varepsilon_1)$, *where* $M = \max\limits_{\mathbf{e}' \in \mathcal{E}(\mathcal{H})} \| \mathbf{e}' \|_2$.

*Proof* Let $\mathbf{e}' \in \mathcal{E}\left(\mathcal{H}\right)$. By writing $\left\langle \mathbf{g}, \mathbf{e}'\right\rangle = \left\langle \mathbf{g} - \mathbf{a}^\star, \mathbf{e}'\right\rangle + \left\langle \mathbf{a}^\star, \mathbf{e}'\right\rangle$ and applying Cauchy-Schwarz inequality to $\left\langle \mathbf{g} - \mathbf{a}^\star, \mathbf{e}'\right\rangle$ we get $\left\langle \mathbf{g}, \mathbf{e}'\right\rangle \leq \left\langle \mathbf{a}^\star, \mathbf{e}'\right\rangle + \varepsilon_0 M$ using condition *(i)*. Consequently

$$\min\limits_{\mathbf{e}' \in \mathcal{E}(\mathcal{H})} \left\langle \mathbf{g}, \mathbf{e}'\right\rangle \leq \min\limits_{\mathbf{e}' \in \mathcal{E}(\mathcal{H})} \left\langle \mathbf{a}^\star, \mathbf{e}'\right\rangle + \varepsilon_0 M = \left\langle \mathbf{a}^\star, \mathbf{e}^\star\right\rangle + \varepsilon_0 M \tag{2}$$

Where the equality is given by Proposition 4. Now, let $\mathbf{e} = \mathbf{E}(h)$, assuming that classifier $h$ satisfies condition *(ii)*. Using $\left\langle \mathbf{a}^\star, \mathbf{e}\right\rangle = \left\langle \mathbf{a}^\star - \mathbf{g}, \mathbf{e}\right\rangle + \left\langle \mathbf{g}, \mathbf{e}\right\rangle$ and Cauchy-Shwarz, we obtain:

$$\left\langle \mathbf{a}^\star, \mathbf{e}\right\rangle \leq \left\langle \mathbf{g}, \mathbf{e}\right\rangle + \varepsilon_0 M \leq \min\limits_{\mathbf{e}' \in \mathcal{E}(\mathcal{H})} \left\langle \mathbf{g}, \mathbf{e}'\right\rangle + \varepsilon_1 + \varepsilon_0 M \leq \left\langle \mathbf{a}^\star, \mathbf{e}^\star\right\rangle + \varepsilon_1 + 2\varepsilon_0 M ,$$

where the first inequality comes from condition *(ii)* and the second inequality comes from (2). The final result is obtained by plugging this inequality into (1). $\quad\square$

Before discussing this result, we first give explicit values of $\mathbf{a}$ and $\Phi$ for pseudo-linear $F$-measures:

**Proposition 6** $F_\beta$, $mF_\beta$ and $mcF_\beta$ defined in Section 2 satisfy the conditions of Proposition 5 with:

$(binary)$ $F_\beta$:
$$\Phi = \frac{1}{\beta^2 P_1} \qquad and \quad \mathbf{a} : t \in [0,1] \mapsto (1+\beta^2 - t, t, 0, 0) \,.$$

$(multilabel\text{--}micro)$ $mF_\beta$:
$$\Phi = \frac{1}{\beta^2 \sum_{k=1}^{L} P_k} \quad and \quad a_i(t) = \begin{cases} 1 + \beta^2 - t & \text{if } i \text{ is odd} \\ t & \text{if } i \text{ is even} \end{cases} \,.$$

$(multiclass\text{--}micro)$ $mcF_\beta$:
$$\Phi = \frac{1}{\beta^2(1 - P_1)} \quad and \quad a_i(t) = \begin{cases} 1 + \beta^2 - t & \text{if } i \text{ is odd and } i \neq 1 \\ t & \text{if } i = 1 \\ 0 & \text{otherwise} \end{cases} \,.$$

*Proof* We prove the result for the $F_\beta$-measure in binary classification. The other cases are similar.

Since $F_\beta$ is fractional-linear as a function of the error profile, it is pseudo-linear on the open convex set $\{\mathbf{e} \in \mathbb{R}^d | (1+\beta^2)P_1 - e_1 + e_2 > 0\}$ (i.e. when the denominator is strictly positive). Moreover, for every set of classifiers $\mathcal{H}$, we have $\mathcal{E}(\mathcal{H}) \subseteq \mathcal{D}_0 = [O, P_1] \times [0, 1 - P_1] \times [1 - P_1] \times [1, P_1]$.

Now, by the definition of $F_\beta$, we have

$$\forall \mathbf{e} \in \mathcal{D}_0, F_\beta(\mathbf{e}) \leq t \quad \Leftrightarrow \quad (1+\beta^2 - t)e_1 + te_2 + (1+\beta^2)P_1(t-1) \geq 0 \,,$$

and the equation still holds by reversing the inequalities. We thus have that:
$\mathbf{a}(t) = (1+\beta^2 - t, t, 0, 0)$ satisfy the condition of Theorem 1 (with $b(t) = (1+\beta^2)P_1(t-1)$).

We now show that the condition of Equation 1 is satisfied for $\mathbf{a}(t) = (1+\beta^2 - t, t, 0, 0)$ and all $\mathbf{e}, \mathbf{e}' \in \mathcal{D}_0$ by taking $\Phi = \frac{1}{\beta^2 P_1}$. To that end, let $\mathbf{e}$ and $\mathbf{e}'$ in $\mathcal{E}(\mathcal{H})$ and $t$ and $t'$ in $\mathbb{R}$ such that $t' = F_\beta(\mathbf{e}') > F_\beta(\mathbf{e}) = t$. Denote by $\varepsilon$ the quantity $\langle \mathbf{a}(t'), \mathbf{e} - \mathbf{e}' \rangle$. Note that $\varepsilon > 0$ and that:

$$
\begin{aligned}
0 &= \langle \mathbf{a}(t), \mathbf{e} \rangle &+ \quad b(t) &= (1+\beta^2 - t)e_1 &+ \quad te_2 &+ \quad (1+\beta^2)P_1(t-1) \\
0 &= \langle \mathbf{a}(t'), \mathbf{e}' \rangle &+ \quad b(t') &= (1+\beta^2 - t')e_1' &+ \quad t'e_2' &+ \quad (1+\beta^2)P_1(t'-1) \\
\varepsilon &= \langle \mathbf{a}(t'), \mathbf{e} - \mathbf{e}' \rangle & &= (1+\beta^2 - t')e_1 &+ \quad t'e_2 &+ \quad (1+\beta^2)P_1(t'-1)
\end{aligned}
$$

where the two first equalities are given by the definitions of $\mathbf{a}$ and $b$, and the last one is obtained using $\langle \mathbf{a}(t'), \mathbf{e} - \mathbf{e}' \rangle = \langle \mathbf{a}(t'), \mathbf{e} \rangle + b(t') - \langle \mathbf{a}(t'), \mathbf{e}' \rangle - b(t')$ and using the second equality.

Taking the difference of the third equality and the first, we obtain:

$$\varepsilon = (t - t')e_1 + (t' - t)e_2 + (1+\beta^2)P_1(t' - t)$$

From which we get, since $(1+\beta^2)P_1 - e_1 + e_2 > 0$ for $\mathbf{e} \in \mathcal{D}_0$:

$$F_\beta(\mathbf{e}') - F_\beta(\mathbf{e}) = t' - t = \varepsilon\big((1+\beta^2)P_1 - e_1 + e_2\big)^{-1} \leq \frac{\varepsilon}{\beta^2 P_1} \,,$$

because $\beta^2 P_1$ the minimum of $(1+\beta^2)P_1 - e_1 + e_2$ on $\mathcal{D}_0$ (taking $e_1 = P_1$ and $e_2 = 0$). We obtain the result since $\varepsilon = \langle \mathbf{a}(t'), \mathbf{e} - \mathbf{e}' \rangle$ by definition. $\square$