[Reviews · NeurIPS 2014]

Submitted by Assigned_Reviewer_1

This paper provides a very solid theoretical analysis of how F-measure and related metrics differ from accuracy-based metrics. While accuracy and cost-based variants are linear functions of true and false positive rates, F-measures are ratios of linear functions. The paper shows that these can be optimised by cost-sensitive classification wrapped in an outer loop which iterates through a range of approximate cost parameters. The main theoretical result gives an approximation guarantee. Some experimental results are reported.

The main strength of the paper is that the analysis is very general, covering a range of F-measures in single-label and multi-label classification and covering both sample and population scenarios. The introduction of error profiles is a nice touch. The main results are sound.

The authors chose to include an experimental section to illustrate the approach and derive some additional conclusions, but this is the weakest part of the paper. It is very dense, doesn't explain some terminology or use it inconsistently, and some results seem to undercut the theoretical analysis ("Although or theoretical developments do not indicate any need to threshold the scores of classifiers, the practical benefits of a post-hoc adjustment of thee scores can be important ..."). For example, the abstract states that the experiments "illustrate the relative importance of weighting and thresholding" but I couldn't find any reference to weighting in Section 4. This section seems to be written in a hurry and would benefit from a much more systematic presentation and also needs to fill some gaps, e.g. provide results for SVM/T in analogy with LR/T.

Further comments, typos etc:

p.1 "training independent classifier" → ... classifiers

p.2 "finite dimensional" → finite-dimensional

p.3 "X x L → {1, 2}" → ... {0.1}

p.4 "multillabel"

p.6 "optimization pseudo-linear" → optimization of ...

p.6 "instanciated"

p.6 Footnote 1 should have an initial capital.

p.7 "We drawn"

p.7 Footnotes have spurious {} in URLs, better formatted with \url{} from \usepackage{url}.

p.8 "is helps"
Summary: Good theoretical work on how F-measure optimisation relates to cost-sensitive classification; experimental section needs work but this isn't a major issue.

Submitted by Assigned_Reviewer_20

The paper concerns a problem of training classifiers that maximize the F-measure. The authors show a reduction of the problem to cost-sensitive classification with unknown costs. They prove that such an approach is consistent and that in practical situations a good approximation can be obtained. The resulting algorithm follows an idea of training several cost-sensitive classifiers with different costs and choosing the one with the best F-measure. This is a quite different approach in comparison with the common solution in which a threshold maximizing the F-measure is tuned on a validation set.

I like this paper, however, I have to admit that the structure of the paper could be improved. For example, I would rather prefer to present first the algorithm and its theoretical analysis for binary classification. Then, the authors could mention the extensions for other settings. I also tried to follow the proofs, but it was not always easy to do that. The authors should consider to move them to an appendix and present in a clearer way.

The authors should also discuss wider the difference between the EUM (expected utility maximization) and DTA (decision-theoretic approach) frameworks, as these two are two different formulations of the problem. For example, the solution for DTA does not necessarily converge to the solution of EUM for the micro F-measure.

The introduced algorithm is very simple. Once we know that the optimal strategy is to find a threshold on conditional probabilities, we can either try to find the threshold on a validation set or use reduction to cost-sensitive classification with different costs (as in reduction of probability estimation to cost-sensitive binary classification). I would rather expect a more sophisticated strategy for finding the cost-sensitive problem with the best F-measure.

Minor comments:
- The sentence: "the optimal classifier (...) for the pseudo-linear F-measures is also an optimal classier for a cost-sensitive classification problem with label dependent costs" should be, in my opinion, stated in the other direction, i.e., "the optimal classifier for a cost-sensitive classification problem with label dependent costs is also optimal for the F-measure". The proof is in the right direction.
- The authors should also present running times of the algorithms.

After rebuttal:

I thank the authors for their responses. I am looking forward for the revised version of the paper.

Summary: This is an interesting paper that advocates an alternative algorithm for optimizing the F-measure. The presentation of the paper could be improved, however, it should be considered for publication at NIPS.

Submitted by Assigned_Reviewer_41

The paper presents a theoretical analysis of F-measure for binary, multiclass and multilabel classification, and an algorithmic approach for optimizing F-measure (approximately) by a reduction to cost-sensitive classification. Existing theoretical results are asymptotic in nature and do not apply for arbitrary function classes. Also, most existing algorithmic approaches to optimizing F-measure solve the problem by thresholding the outputs of standard classifiers. Experiments show that optimizing F-measure by a reduction to cost-sensitive classification can be preferred over thresholding based methods as well as structured prediction methods.

This is not my area of expertise, and so it is difficult for me to judge the significance of the theoretical results when compared to existing literature. Having said this, I find the contribution to be strong. There is an interesting mix of theory, algorithms and experiments in the paper. The algorithmic approach that uses a reduction to a much simpler problem is particularly interesting from a practical standpoint. The paper is well-written and mostly self-contained.

One notable gap in the proposed framework is that it cannot be used to optimize the per-instance F-measure in multilabel classification, which I believe is a natural function to optimize for multilabel classification. Practically efficient method, via reduction to binary classification, to optimize this function exactly exists in the literature [6].
Summary: Strong contribution, with an interesting mix of theory, algorithms and experiments.
Author Feedback
Author rebuttal: We thank the reviewers for their careful reading, and for their suggestions and comments; we will work out the presentation and the discussions:
- improve the writing of the experimental section as suggested by AR1,
- add a paragraph describing the algorithm and the results on binary classification early in the analysis section, improve discussions on EUM vs DTA and on the oppotunity of improving the strategy for finding the optimal costs as suggested by AR20,
- improve the discussion on the fact that our Framework does not capture the per-instance F1 as suggested by AR41.

specific answers:

- AR1: when we used the term "weighting" in the abstract, we intended to refer to varying the label-dependent costs. We will correct this inconsistency of terminology.

- AR20: we will add the running times of the algorithms. The overall running time is heavily dependent on the implementation of the cost-sensitive algorithms and the stopping criteria that are used. In Liblinear (the implementation we use), logistic regression is extremely fast even with a very "conservative" stopping criterion; SVMs with a hinge loss can be slower by a factor as high as 50 (for large values of C) to achieve sufficient convergence (average 10 times on our range of C).